# Development and Clinical Trial of a New Orthopedic Surgical Robot for Positioning and Navigation

**DOI:** 10.3390/jcm11237091

**Published:** 2022-11-30

**Authors:** Jie Chang, Lipeng Yu, Qingqing Li, Boyao Wang, Lei Yang, Min Cheng, Feng Wang, Long Zhang, Lei Chen, Kun Li, Liang Liang, Wei Zhou, Weihua Cai, Yongxin Ren, Zhiyi Hu, Zhenfei Huang, Tao Sui, Jin Fan, Junwen Wang, Bo Li, Xiaojian Cao, Guoyong Yin

**Affiliations:** 1The First Affiliated Hospital of Nanjing Medical University, Nanjing 210029, China; 2The Second Affiliated Hospital of Nanjing Medical University, Nanjing 210003, China; 3Department of Orthopedics, The Affiliated Taizhou People’s Hospital of Nanjing Medical University, Taizhou School of Clinical Medicine, Nanjing Medical University, Taizhou 225300, China; 4Nanjing Tuodao Medical Technology Co., Ltd., Nanjing 210012, China; 5Wuhan Fourth Hospital, Puai Hospital, Tongji Medical College, Huazhong University of Science and Technology, Wuhan 430030, China; 6Guizhou Provincial People’s Hospital, Guiyang 550002, China

**Keywords:** pedicle screw placement, robot assisted, randomized controlled trial, positioning and navigation, 3D fluoroscopy, spine surgery

## Abstract

Robot-assisted orthopedic surgery has great application prospects, and the accuracy of the robot is the key to its overall performance. The aim of this study was to develop a new orthopedic surgical robot to assist in spinal surgeries and to compare its feasibility and accuracy with the existing orthopedic robot. A new type of high-precision orthopedic surgical robot (Tuoshou) was developed. A multicenter, randomized controlled trial was carried out to compare the Tuoshou with the TiRobot (TINAVI Medical Technologies Co., Ltd., Beijing) to evaluate the accuracy and safety of their navigation and positioning. A total of 112 patients were randomized, and 108 patients completed the study. The position deviation of the Kirschner wire placement in the Tuoshou group was smaller than that in the TiRobot group (*p* = 0.014). The Tuoshou group was better than the TiRobot group in terms of the pedicle screw insertion accuracy (*p* = 0.016) and entry point deviation (*p* < 0.001). No differences were observed in endpoint deviation (*p* = 0.170), axial deviation (*p* = 0.170), sagittal deviation (*p* = 0.324), and spatial deviation (*p* = 0.299). There was no difference in security indicators. The new orthopedic surgical robot was highly accurate and optimized for clinical practice, making it suitable for clinical application.

## 1. Introduction

In recent years, more patients have needed to undergo pedicle screw fixation due to a variety of spinal pathologies [1,2], and screw implantation accuracy is critical. As an auxiliary medical device, surgical robots can improve the accuracy and safety of surgery and are increasingly favored by doctors [3,4]. For example, the Da Vinci robot has been widely used in general surgery [5], urology [6], gynecology [7], and other fields. Robot-assisted spinal surgery has also been applied in clinical practice [8]. It has been confirmed to be superior to manual operations in terms of safety, accuracy, the amount of radiation to surgeons, and the learning curve [9,10].

At present, many spinal surgery robots have been developed in the world, such as Mazor X^®^ (Mazor Robotics Ltd., Caesarea, Israel), ROSA^®^ SPINE (Zimmer Biomet Robotics, Montpellier, France), and Excelsius GPS^®^ (Globus Medical, Inc., Audubon, PA, USA) [11]. The TiRobot system (TINAVI Medical Technologies Co., Ltd., Beijing, China) is the first orthopedic robot created entirely in China. It was approved by the China Food and Drug Administration in 2016. The TiRobot device has a tracking-capable robotic arm combined with an intraoperative three-dimensional (3D) navigation system [12,13]. After acquiring images preoperatively and planning the desired screw trajectory, the surgeon manually performs drilling and screw insertion. However, the TiRobot has some shortcomings, including trauma caused by its tracker, and slightly complicated operation procedures. Therefore, our team independently developed a new orthopedic robot (Tuoshou, Nanjing Tuodao Medical Technology Co., Ltd., Nanjing, China) to address the challenges associated with the existing robots.

The present study also verified the accuracy of the robot. In the early stages, animal experiments confirmed the accuracy of the robot in order to meet the requirements of pedicle screw placement. A multicenter, randomized controlled trial (RCT) was then conducted to compare the Tuoshou with the TiRobot. The primary objective of this RCT was to investigate the differences between the two robots in the accuracy of Kirschner wire (K-wire) placement during pedicle screw fixation. Other accuracy and safety metrics were also compared.

## 2. Materials and Methods

### 2.1. Study Design

After obtaining the institutional review board approval, a multicenter RCT comparing the new robot to the TiRobot in terms of pedicle screw fixation was performed. The trial followed the Consolidated Standards of Reporting of Trials (CONSORT) guidelines [14] and was carried out in accordance with the principles of the Declaration of Helsinki. It was registered on clinicaltrials.gov (NCT04883580). Participating centers included The First Affiliated Hospital of Nanjing Medical University, Guizhou Provincial People’s Hospital, and Wuhan Fourth Hospital.

### 2.2. Participants

Selection criteria included age of 18–75 years and internal fixation of the thoracic or lumbar pedicle screw performed based on the patient’s lesion segment. All patients signed the informed consent form and voluntarily received treatment and follow-up according to the requirements of the experimental protocol.

### 2.3. Robot Components

The robot consists of a robotic base station, an optical tracking system (OTS), and a toolset for navigation and positioning, together with surgical navigation and positioning software (Figure 1 and Figure 2). First, the robot’s C-shaped arm captures the image of the vertebrae and transmits the image data back to the robotic base station. Then the OTS tracks the calibrator and the tracker. The calibrator performs tracker registration against the image, establishing a correlation between the two, such that any subsequent change in the position of the vertebrae corresponds to a change in the position of the tracker. The surgeon then plans the surgical pathway on the image displayed on the robotic base station. Next, the system plans the movements of the robotic arm based on this pathway, tracking the position of the robotic arm’s end-effector via the OTS to compensate for its movement in real-time. This ensures that it reaches the position planned by the surgeon, indicating the direction the surgeon must follow through the cannula to establish the pathway, thus helping the surgeon to complete operations, such as leading and inserting guidewires, drilling, and screw placement.

### 2.4. Interventions

Patients in the Tuoshou group were placed in a prone position on the operating table after general anesthesia administration and were prepared and draped under sterile conditions. After confirming the surgical vertebral body position using a C-arm scanner, the K-wire was placed percutaneously into the spinous process of the adjacent vertebral body and attached to the fixator. The purpose of the fixator was to secure the tracker to the patient. The calibrator was held by the robotic arm and placed close to the operating site. After a 3D scan with a C-arm scanner, the surgeon was able to plan the screw trajectories in axial, coronal, and sagittal views and select the optimal size of the screw on the console. Then, the robotic arm moved to the specified position according to the set trajectory to indicate the direction of screw implantation. Furthermore, surgeons drilled through the K-wire into the pedicle under real-time navigation monitoring. The screw was then inserted into the pedicle based on the K-wire guidance. Decompression and cage placement procedures were performed next as needed (Figure 3).

The TiRobot’s surgical procedure was similar to that of the Tuoshou. In addition to the different planning methods of the console, the spinous process needs to be exposed when fixing the reference tracker in the TiRobot group, while there was no big cut created when fixing the reference tracker in the Tuoshou robot group [15].

### 2.5. Randomization and Quality Control

Doctors evaluated patients in outpatient clinics and asked them if they would join the robotics trial. Patients were given at least one week to consider their participation in the trial and provide their informed consent for inclusion. Patients who signed the informed consent form were randomized 1:1 via Interactive Web Response System (IWRS) and assigned to the experimental or control groups on the day of surgery (Figure 4). All surgeries were performed by the same experienced surgeons at each trial center. Six surgeons were included in this clinical trial and each surgeon performed at least 50 Tuoshou-assisted and 50 TiRobot-assisted pedicle screw fixation procedures. All personnel participating in the clinical trial were required to follow regulations, undergo protocol training, and understand regulatory requirements and protocol content. Each trial center conducted detailed training for its research participants, including the specification for the informed consent process, inclusion and exclusion criteria, trial procedures, use of trial products, documentation requirements, and reporting of adverse events and protocol deviations.

### 2.6. Outcomes

The primary outcome of this trial was the deviation of the K-wire placement position, that is, the deviation between the computer-planned position and the actual position. Secondary outcomes included pedicle screw insertion accuracy rate (Gertzbein--Robbins classification) [16], entry point deviation, endpoint deviation, axial deviation, sagittal deviation, and spatial deviation. As for the pedicle screw insertion accuracy rate, grades A and B screw positions were considered clinically acceptable. The entry point and endpoint deviation, respectively, referred to the entry point of the planned position and the spatial distance from the axis to the actual insertion position of the K-wire. Axial and sagittal plane angle deviation refers to the minimum angle between the planned position on the axial plane and the sagittal plane and the projected position of the K-wire, respectively. Spatial angle deviation refers to the minimum angle between the planned position in space and the actual position of the K-wire.

The operation time was defined as the time from the start of the orthopedic surgical system registration to the end of screw placement. Instrument success means that during the procedure, the operator completed the surgical channel planning based on the 3D scanning image data, and the arm of the orthopedic surgery system moved to the position specified in the surgical plan without instrument error. Technology success refers to the successful placement of K-wires and screws along the guide and sleeve during the operation without the need to reposition the screws or switch to freehand screw placement. Operation success was defined as the success of instruments and technology during the operation without the occurrence of serious surgical complications, such as nerve root injury, vascular injury, spinal cord injury, visceral injury, and pedicle fracture. All postoperative adverse events were recorded at seven-day follow-up, and the incidence of adverse events was compared between the two groups.

### 2.7. Sample Size

The main efficacy index of the present study was the position deviation of the K-wire placement. According to previous literature reports, the deviation of pedicle screw placement in orthopedic surgery navigation and positioning of the system-assisted pedicle screw internal fixation can reach the level of 1.77 ± 0.78 mm. It was assumed that the deviation of the insertion position of the K-wires in the experimental and control groups was 1.5 mm, the combined standard deviation of the two groups was estimated to be 1 mm, and the non-inferiority margin was 0.6 mm. The sample size was 45 cases in each group, and the dropout rate was preset to 20%. The final calculation resulted in a sample size of 112 cases.

#### Statistical Analysis

The datasets were divided into three categories: full analysis set (FAS), per-protocol set (PPS), and safety set (SS). The FAS refers to the set of subjects determined according to the principle of intention-to-treat, which is a data set composed of all subjects who participated in randomization and used the research product. The PPS refers to all of the treatment population subgroups that have completed the trial without serious violations of the protocol (referring to patients violating the inclusion or exclusion criteria) or poor compliance. The SS refers to the data set consisting of all subjects who participated in the random use of the investigational product and underwent a safety evaluation. Efficacy analysis was performed on the FAS and PPS, baseline demographic data were collected based on the FAS, and evaluation of safety data was performed on the SS.

The comparison of the two indicator groups was analyzed using appropriate methods according to the indicator type. Group comparison of quantitative data used a group *t*-test (homogeneity of variance, normal distribution) or Mann--Whitney U test according to the data distribution. The Chi-square test or exact probability method (if the Chi-square test was not applicable) was used for categorical data. Rank data were analyzed via the Mann--Whitney U test or CMH (Cochran--Mantel--Haenszel) test. All statistical tests were two-sided, and a *p*-value of <0.05 was considered statistically significant for the differences tested.

## 3. Results

### 3.1. Description of the Sample

A total of 114 patients requiring pedicle screw implantation between April 2021 and October 2021 were eligible for the trial. As a result, 112 patients were included in the trial, with 56 each in the TiRobot and Tuoshou groups. One patient in the Tuoshou group voluntarily withdrew from the trial due to high cardiovascular risk, and another patient refused surgery. One patient in the TiRobot group was deemed unsuitable for surgical treatment with an orthopedic robot, and another patient was deemed unsuitable for surgical treatment (Figure 4). Overall, there were 54 patients in each group, which met the sample size requirement. Patients with spinal trauma and degenerative spinal disease were the main components, including some patients with intraspinal tumors. Demographic and clinical characteristics were similar at baseline (Table 1 and Table 2). There was no statistical difference between the two groups in the number of screws and vertebral bodies (Table 3).

### 3.2. Outcomes

In the FAS, one patient in the TiRobot group had missing data for L5 bilateral K-wire placement and was replaced with the worst result of all patients in the FAS (5.683 mm). The missing data in the PPS was not included in the analysis. In both datasets, the position deviation of K-wire placement in the Tuoshou group was significantly smaller than that in the TiRobot group (Table 4). The mean difference in the position deviation of K-wire placement between the two groups was −0.24 mm with a two-sided 95% confident interval (CI) of mean difference being −0.47 to −0.08 in the FAS and −0.22 mm with two-sided 95% CI of mean difference being 0.38 to −0.05 in the PPS.

The pedicle screw insertion accuracy rate was 98% in the Tuoshou group and 94% in the TiRobot group. The entry point deviation of the Tuoshou group was significantly smaller than that of the TiRobot group, while the endpoint deviation between the two groups was not statistically significant. In addition, there was no significant difference in axial deviation, sagittal deviation, and spatial deviation between the two groups (Table 4).

### 3.3. Security Indicators

Safety evaluations were performed using the SS (Table 5). There was no significant difference in the effect of the two surgical robots on the operation time. Success rate outcomes also did not differ in the instrument, technology, and operation between the groups. The rate of postoperative adverse events was 37% in the Tuoshou group and 39% in the TiRobot group, and each specific adverse event was listed in the study (Table 6). Only one patient in each group needed a blood transfusion or special drug treatment (Clavien--Dindo II). There was no significant difference in the incidence of postoperative adverse events between the two groups.

## 4. Discussion

The orthopedic surgical robot has the advantages of safety, accuracy, less radiation, and a shorter learning curve, and more and more surgeons have begun to perform robot-assisted surgeries [17]. In spinal surgeries, robot assistance is often adopted for pedicle screw insertion, navigation, and positioning in vertebroplasty [18,19,20,21,22]. Therefore, the accuracy of robot navigation and positioning directly affects the quality of spinal surgeries.

Currently, the TiRobot is the most commonly used among the existing orthopedic surgical robots in China. Although it has improved a lot in orthopedic surgeries, there are still some disadvantages associated with it. First, the TiRobot consists of three main components, including a robotic arm, an optical tracking device, and a surgical planning and controlling workstation [15]. The relatively large number of components affects the efficiency of the operation. The small operating room space could affect the surgeon’s operation. In some cases, the operator may not be able to get accurate real-time feedback through the display screen while inserting the K-wire. The Tuoshou integrates the robotic arm and the workstation, which greatly improves surgical efficiency and space utilization. The operator can obtain the positional error information of the robotic arm and the K-wire through the screen at any time, which improves the manipulability of the robot.

Second, the end of the robotic arm in the Tuoshou has been improved compared to that of the TiRobot, so that the accuracy of surgical positioning can be guaranteed. A tracker was installed on the end device of the robotic arm to assist in visual positioning. The ruler for registration and the guide cylinder for orientation were connected to the end through disassembly and assembly mechanisms. Repeated disassembly and assembly for a long period of time will lead to inconsistency of physical wear and installation position, resulting in loss of accuracy. On the contrary, the end of the Tuoshou adopts the integrated design of the tracker and guide cylinder, which avoids the risk of accuracy decline caused by disassembly and assembly.

The third advantage of the Tuoshou compared to the TiRobot is less surgical trauma. In the TiRobot group, the fixation of the reference tracker was achieved after the skin was incised and fixed on the spinous process. Sometimes the incision made by this operation was not smaller than the incision made by the screw placement, which defeats the purpose of minimally invasive surgery. Using our fixation device with the K-wire firmly fixed the tracer on the spinous process without exposure, which greatly reduced trauma.

An animal experiment on eight sheep was conducted before the Tuoshou was used for pedicle screw fixation in a human. The results showed that the Tuoshou has a current mean deviation of 0.839 mm, which is already superior to that of commercially available surgical robots. Furthermore, based on the present randomized trial, patients had good clinical and patient-reported outcomes after pedicle screw fixation, both assisted by the TiRobot and Tuoshou. When the patients’ baselines were similar, the K-wire placement deviation in the Tuoshou group was significantly smaller than that in the TiRobotic group [(1.81 ± 0.89) mm vs. (2.02 ± 1.05) mm, *p* = 0.021]. Pedicle screw insertion accuracy rate and entry point deviation, two measures of response accuracy, were smaller and statistically different in the Tuoshou group.

Since our study only involved Chinese robots, we made a preliminary comparison with other robots. The number rate of screws reaching class A and B of the Gertzbein--Robbins scale in the robot group is reported to be 91–99% [23,24,25,26], which is similar to the results of the Tuoshou robot. In terms of accuracy, the robot has reached the level of the world’s mainstream spinal surgery robot. Furthermore, few complications happened in the Tuoshou group, which is consistent with the high safety of robotic surgery. Additionally, similar to other spinal surgical robots, the Tuoshou robot only needs one time of three-dimensional scan during the operation, which greatly reduces radiation exposure [27,28].

There were some limitations in this study. First, the purpose of this investigation was to evaluate the accuracy of the robot in pedicle screw fixation, so the disease type of the enrolled patients was not specified. Second, the follow-up time was seven days after surgery, which was relatively short, and indicators, such as pain improvement and subsequent imaging changes, were not evaluated in the study. Third, although this was a multicenter trial, only surgeons who are experts in minimally invasive repair were included, which may limit its generalizability to surgeons with less experience. Additionally, the TiRobot was used in the control group for the study, which is commonly used in China. Other commonly used orthopedic robots in the world, such as the Mazor Robot [29], were not included, which could lead to the one-sidedness of the test results.

## 5. Conclusions

In this study, we verified whether the robot we developed could be used in spinal surgery. Compared with the existing orthopedic robot, the deviation between the computer-planned position and the actual position was smaller, demonstrating its high accuracy in pedicle screw implantation.

## Figures and Tables

**Figure 1 jcm-11-07091-f001:**
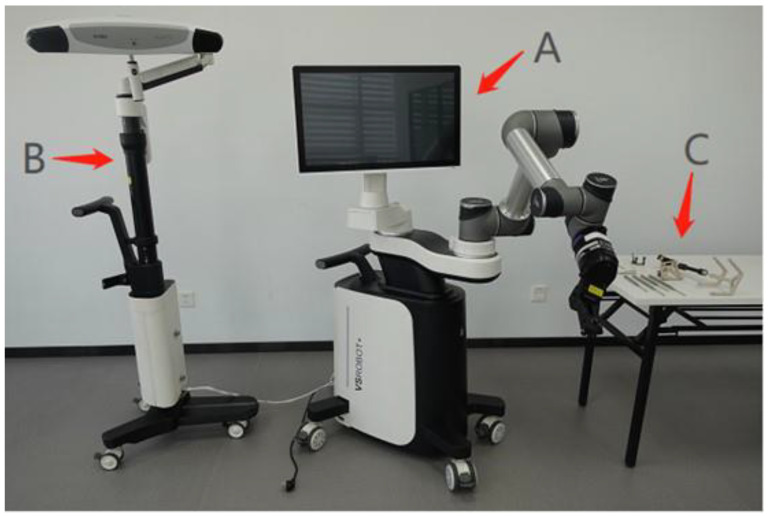
Robotic base station (A), optical tracking system (B), and toolset (C).

**Figure 2 jcm-11-07091-f002:**
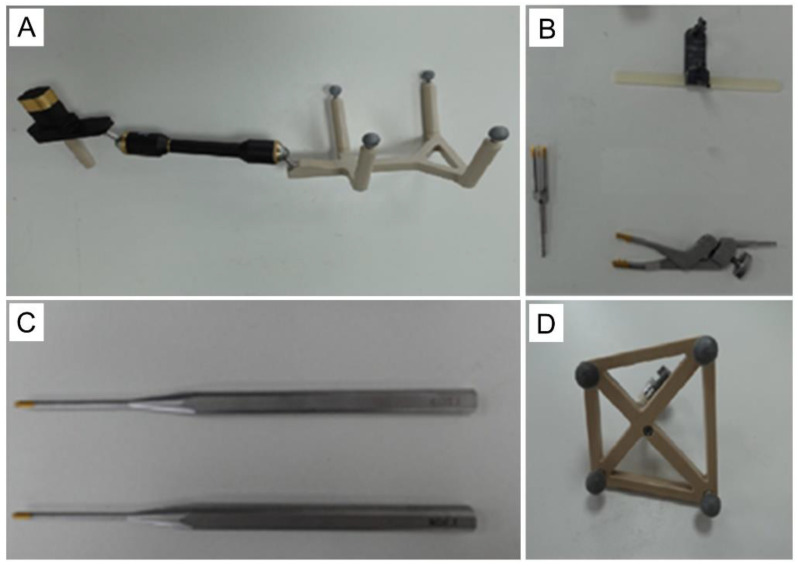
The toolset of the Tuoshou robot. (**A**) Calibrator. (**B**) Connectors. (**C**) Cannula. (**D**) Tracker.

**Figure 3 jcm-11-07091-f003:**
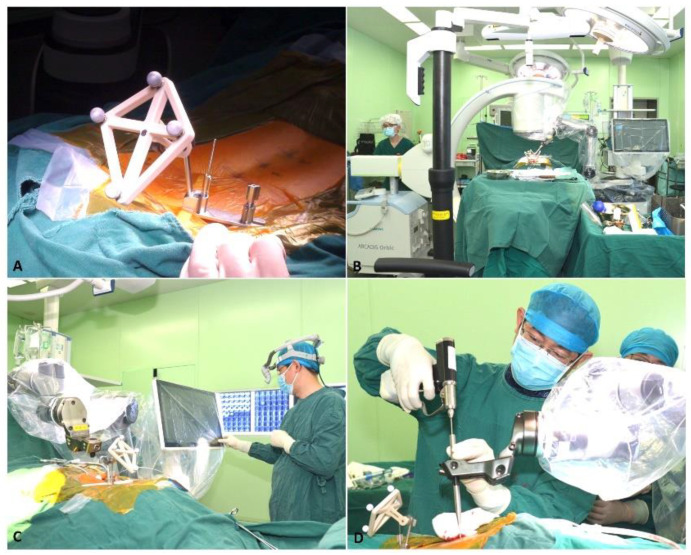
Securing the tracker (**A**), scanning (**B**), planning (**C**), and drilling the K-wire (**D**).

**Figure 4 jcm-11-07091-f004:**
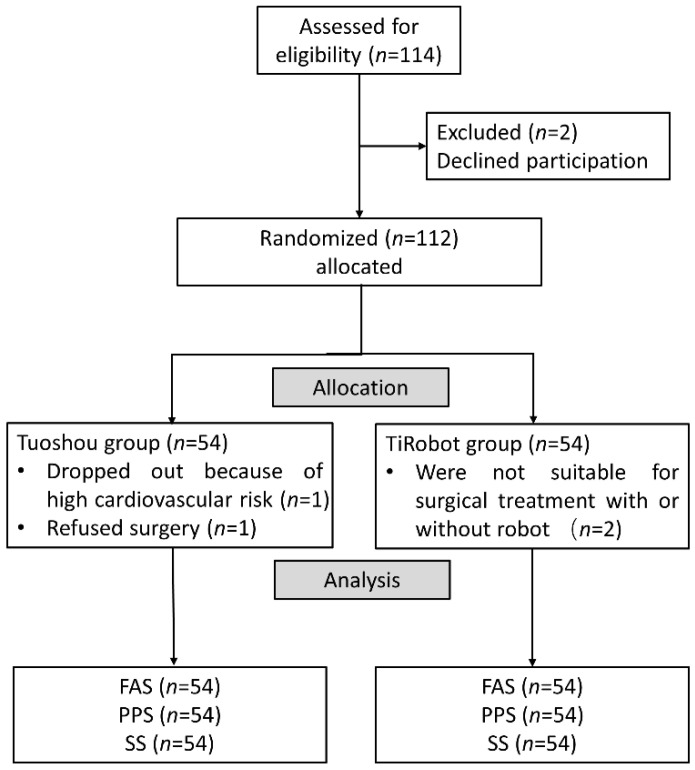
Flow chart of included patients.

**Table 1 jcm-11-07091-t001:** Baseline Characteristics.

Variable	Tuoshou (*n* = 54)	TiRobot (*n* = 54)	*p*-Value
Mean age, years (IQR)	55.50 (50.00 to 66.00)	57.00 (50.00 to 64.00)	0.796 *
Mean height, cm (IQR)	163.00 (160.00 to170.00)	165.00 (159.00 to 170.00)	0.966 *
Mean weight, kg (IQR)	65.00 (60.00 to 74.00)	62.75 (58.00 to 68.00)	0.123 *
Mean BMI, kg/m^2^ (SD)	24.74 (3.48)	23.53 (2.67)	0.045 ^†^
Sex, n (%)			0.845 ^‡^
Male	32 (59)	31 (57)	
Female	22 (41)	23 (43)	

* Mann-Whitney U test. ^†^ Independent- samples *t*-test. ^‡^ Chi-square test. SD, standard deviation; BMI, body mass index; IQR, interquartile range.

**Table 2 jcm-11-07091-t002:** Blood test results of two groups.

Variable	Tuoshou, *n* (%)	TiRobot, *n* (%)	*p*-Value
White blood cell count			0.448 *
Normal	46 (85)	43 (80)	
Abnormal	8 (19)	11 (20)	
Red blood cell count			0.820 *
Normal	42 (78)	41 (76)	
Abnormal	12 (22)	13 (24)	
Platelet count			0.567 *
Normal	48 (89)	46 (85)	
Abnormal	6 (11)	8 (15)	
Alanine aminotransferase			1.000 *
Normal	49 (91)	49 (91)	
Abnormal	5 (9)	5 (9)	
Aspartate aminotransferase			0.267 *
Normal	45 (83)	48 (89)	
Abnormal	9 (17)	5 (9)	
Not tested	0 (0)	1 (2)	
Creatinine			0.740 ^†^
Normal	50 (93)	48 (89)	
Abnormal	4 (7)	6 (11)	
Prothrombin time			0.243 ^‡^
Normal	54 (100)	51 (94)	
Abnormal	0 (0)	3 (6)	
Activated partial thromboplastin time			0.037 ^†^
Normal	53 (98)	46 (85)	
Abnormal	1 (2)	8 (15)	

* Chi-square test. ^†^ Yates’ continuity corrected Chi-square test. ^‡^ Fisher’s exact test.

**Table 3 jcm-11-07091-t003:** Number of vertebral bodies and screws.

Variable	Tuoshou	TiRobot	*p*-Value
Number of vertebral bodies, n (%)			0.560 *
1	2 (4)	3 (5)	
2	30 (56)	26 (48)	
3	13 (24)	22 (41)	
4	7 (13)	2 (4)	
5	2 (3)	1 (2)	
Average value (IQR)	2.00 (2.00 to 3.00)	2.00 (2.00 to 3.00)	0.917 ^†^
Number of screws, n (%)			0.590 *
1	0 (0)	0 (0)	
2	2 (4)	3 (6)	
3	0 (0)	0 (0)	
4	30 (55)	26 (48)	
5	0 (0)	1 (2)	
6	14 (26)	22 (42)	
7	1 (2)	0 (0)	
8	5 (9)	1 (2)	
9	0 (0)	0 (0)	
10	2 (4)	1 (2)	
Average value (IQR)	4.00 (4.00 to 6.00)	4.00 (4.00 to 6.00)	0.892 ^†^

* Cochran-Mantel-Haenszel test. ^†^ Mann-Whitney U test. IQR, interquartile range.

**Table 4 jcm-11-07091-t004:** Primary and secondary outcomes of FAS and PPS.

	FAS			PPS		
Outcomes	Tuoshou	TiRobot	*p*-Value	Tuoshou	TiRobot	*p*-Value
Primary outcome						
Number of K-wire (Missing)	275 (0)	265 (0)	N/A	275 (0)	263 (0)	N/A
Average value, mm (IQR)	1.66 (1.14 to 2.31)	1.89 (1.18 to 2.62)	0.014 *	1.66 (1.14 to 2.31)	1.88 (1.18 to 2.62)	0.021 *
Secondary outcomes						
Pedicle screw insertion accuracy rate, n (%)	271 (98)	250 (94)	0.016 ^†^	271 (98)	250 (94)	0.016 ^†^
Entry point deviation, mm (IQR)	1.55 (0.93 to 2.09)	1.88 (1.20 to 3.12)	<0.001 *	1.55 (0.93 to 2.09)	1.88 (1.20 to 3.12)	<0.001 *
Endpoint deviation, mm (IQR)	1.77 (1.22 to 2.60)	1.70 (1.07 to 2.40)	0.225 *	1.77 (1.22 to 2.60)	1.70 (1.07 to 2.40)	0.170 *
Axial deviation, ° (IQR)	0.72 (0.26 to 1.35)	0.69 (0.16 to 1.67)	0.892 *	0.72 (0.26 to 1.35)	0.69 (0.16 to 1.67)	0.988 *
Sagittal deviation, ° (IQR)	0.99 (0.51 to 1.84)	1.18 (0.54 to 2.02)	0.258 *	0.99 (0.51 to 1.84)	1.18 (0.54 to 2.02)	0.324 *
Spatial deviation, ° (IQR)	1.55 (0.91 to 2.43)	1.57 (0.93 to 2.70)	0.238 *	1.55 (0.91 to 2.43)	1.57 (0.93 to 2.70)	0.299 *

* Mann-Whitney U test. ^†^ Yates’ continuity corrected Chi-square test. IQR, interquartile range.

**Table 5 jcm-11-07091-t005:** Differences between two groups in security indicators.

	SS		
Security Indicators	Tuoshou	TiRobot	*p*-Value
Operation time, min (IQR)	64.50 (55.00 to 80.00)	56.00 (45.00 to 76.00)	0.085 *
Instrument success, n (%)	54 (100)	54 (100)	N/A
Technology success, n (%)	54 (100)	51 (94)	0.243 ^†^
Operation success, n (%)	54 (100)	51 (94)	0.243 ^†^
Postoperative adverse events, n (%)	20 (37)	21 (39)	0.843 ^†^
Clavien-Dindo I/II, n	19/1	20/1	0.490 ^†^

* Chi-square test. ^†^ Yates’ continuity corrected Chi-square test.

**Table 6 jcm-11-07091-t006:** Postoperative adverse events of two groups.

	SS		
Postoperative Adverse Events	Tuoshou, *n* (%)	TiRobot, *n* (%)	*p*-Value
**Total**	20 (37)	21 (39)	0.843 *
**Gastrointestinal system**	5 (9)	6 (11)	0.750 *
Stomach ache	3 (6)	1 (2)	0.610 ^†^
Vomit	1 (2)	3 (6)	0.610 ^†^
Nausea	1 (2)	3 (6)	0.610 ^†^
Bloating	0 (0)	1 (2)	1.000 ^‡^
**Injury**	5 (9)	5 (9)	1.000 *
Bleeding	4 (7)	5 (9)	1.000 ^†^
Incision complications	1 (2)	0 (0)	1.000 ^‡^
**Systemic diseases and drug reactions**	2 (4)	6 (11)	0.270 ^†^
Fever	1 (2)	6 (11)	0.118 ^†^
Chest discomfort	1 (2)	1 (2)	1.000 ^†^
**Neurological complications**	5 (9)	0 (0)	0.057 ^‡^
Cerebrospinal fluid leakage	5 (9)	0 (0)	0.057 ^‡^
**Vascular and lymphatic complications**	3 (5)	2 (4)	1.000 ^†^
Deep vein thrombosis	3 (5)	2 (4)	1.000 ^†^
**Cardiac complications**	2 (4)	1 (2)	1.000 ^†^
Palpitation	1 (2)	1 (2)	1.000 ^†^
Atrial fibrillation	1 (2)	0 (0)	1.000 ^‡^
**Metabolic and nutritional complications**	1 (2)	1 (2)	1.000 ^†^
Hypoproteinemia	1 (2)	1 (2)	1.000 ^†^
**Infection**	0 (0)	1 (2)	1.000 ^‡^
Urinary tract infection	0 (0)	1 (2)	1.000 ^‡^
**Respiratory complications**	1 (2)	0 (0)	1.000 ^‡^
Expectoration	1 (2)	0 (0)	1.000 ^‡^
Cough	1 (2)	0 (0)	1.000 ^‡^
**Skin complications**	0 (0)	1 (2)	1.000 ^‡^
Mild pressure sore (I)	0 (0)	1 (2)	1.000 ^‡^
**Urinary complications**	0 (0)	1 (2)	1.000 ^‡^
Urethral pain	0 (0)	1 (2)	1.000 ^‡^
**Hematological complications**	0 (0)	1 (2)	1.000 ^‡^
Anemia	0 (0)	1 (2)	1.000 ^‡^
**Ocular complications**	1 (2)	0 (0)	1.000 ^‡^
Blurred vision	1 (2)	0 (0)	1.000 ^‡^

* Chi-square test. ^†^ Yates’ continuity corrected Chi-square test. ^‡^ Fisher’s exact test.

## Data Availability

The dataset is not publicly available due to privacy and ethical restrictions. The data for this study are available upon request addressed directly to the Research Ethics Boards of the lead institution, which must first approve the request. If the request is approved, anonymized data supporting the conclusions of this manuscript will be made available by the corresponding author.

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
