# Peer review of "Development and Clinical Trial of a New Orthopedic Surgical Robot for Positioning and Navigation"

_jcm, 2022, doi:10.3390/jcm11237091_

Round 1

Reviewer 1 Report

Congratulations to the authors, apart from 2 adjustments in the introduction section the whole manuscript is well written and scientifically profound. Moreover their work adds to the improvement of robot-assisted surgery in orthopedics.

p1, line 31ff - patients are not stabilized bc of spinal stenosis (laminotomy, laminoplasty etc.) only if you create instability or degenerative disease is also present, rephrase - maybe just generalize "have undergone pedicle screw placement / instrumented dorsal stabilisation due to a variety of spinal pathologies"

line 32 - screw placement accuracy has always been crucial...

Reviewer 2 Report

Thank you for the submission of this manuscript about a new robot for spinal surgery. Before further proceeding of publication act, I suggest to address the following issues:

Introduction:

- page 2, line 50: Can you provide literature about theses early animal experiences?

- general: for the authors not being that much in the robot topic: can you provide a short overview about other robots in spine surgery? Do not only focus on chinese products!

Materials and Methods:

- page 2, line 68: "...performed based on the patients's condition". What does that mean? Please rephrase.

- page 4, line 103: "The difference was..." Please rephrase this sentence. What exactly was the difference?

- page 4, line 112: How many surgeons were included in this study?

- general: What patients are included in the study? Spinal trauma, spinal tumor, spinal degeneration? Scoliosis surgery? Please give an detailed overview.

Results:

- general: The complication report is not adequate for the study purpose. First, I suggest to classify the complications according to severity. You may use the Clavien/Dindo classification. Second, we have to distinguish between relevant complications and non-relvant complications according to the study purpose. Example: A bladder tumor within 7 days after surgery is not relevant when talking about robot complications!

Discussion:

- page 11, line 234: You are publishing in an international journal. Please do not only focus on China. What is with robots produced and used in other countries? What about their performance? Please include internation studies in order to compare your robot to others!

- page 11, line 239: Please rephrase.

- general: In the discussion section, your study data are repeated and TiRobot is compared to Tuoshuo. This is not a valid discussion of study results for an international scientific journal. Please compare your results to international results based on literature.

Conclusion:

- page 12, line 282: This is not a study about developing a robot! Please rephrase!

Figures:

- Figure 1: Please provide a more detailed photograph of the tool set.

- Figure 2: Delete picture A and picture F. Not any important information on the system is given by these pictures. Provide the other pictures in a bigger manner, that details could be presented better.

- Figure 3: The flowsheet is not correct. Other numbers are given in the text. Please clearify:

(114 patients assed) - (2 patients excluded) - (2 patients declined) = 110 patients that could be randomized.

110 patients randomized in two groups = 55 patients per group.

Two dropouts per group results in 53 patients per group for FAS, PPS and SS

Tables:

- Table 2: What information on the study does table 2 provide?

Reviewer 3 Report

First I would like to commend the authors for their quality work and the development of a new surgical robot for spinal applications. The manuscript is well written and concise.

- In Figure 3 : Please replace "flow sheet" by "flow chart" in the title

- Please precise what scale your are using to define "pedicle screw insertion accuracy". The Gertzbein-Robbins scale is the most commonly used, grade A being considered as perfect and B as good pedicle screw placement. This need to be precised and ideally adjusted to the gertzbein-robbins scale.

- Another interesting consideration to have is the superior facet joint sparing. An evaluation scale was developed and proposed by Yson et al. (PMID: 23197012). Did you consider this analysis ?

- in the discussion, I think it may be appropriate to mention the other commercial available robots and the potential place of the Tuoshuo among them interns of benefits and advantages. You may also mention the reduction in radiation exposure for both the patient and the surgical team.
